materials science/nanotechnology/energy

nanoparticle, fabric, infrared, self-heating, moisture management, quick-dry

**Author for correspondence:**
K. M. Nalin De Silva
e-mail: kmnd@chem.cmb.ac.lk

This article has been edited by the Royal Society of Chemistry, including the commissioning, peer review process and editorial aspects up to the point of acceptance.

# Infrared absorbing nanoparticle impregnated self-heating fabrics for significantly improved moisture management under ambient conditions

Lahiru A. Wijenayaka[1,2], Ruchira N. Wijesena[1], Nadeeka D. Tissera[1], W. R. L. Nisansala Bandara[1], Gehan J. Amaratunga[1,3] and K. M. Nalin De Silva[1,4]

[1]Sri Lanka Institute of Nanotechnology (SLINTEC), Mahenwatte, Pitipana, Homagama 10200, Sri Lanka
[2]Department of Chemistry, The Open University of Sri Lanka, Nawala 10250, Sri Lanka
[3]Department of Engineering, University of Cambridge, Cambridge CB3 0HE, UK
[4]Department of Chemistry, University of Colombo, Colombo 03 00300, Sri Lanka

KMNDS, 0000-0003-3219-3233

Propensity of a textile material to evaporate moisture from its surface, commonly referred to as the 'moisture management' ability, is an important characteristic that dictates the applicability of a given textile material in the activewear garment industry. Here, an infrared absorbing nanoparticle impregnated self-heating (IRANISH) fabric is developed by impregnating tin-doped indium oxide (ITO) nanoparticles into a polyester fabric through a facile high-pressure dyeing approach. It is observed that under simulated solar radiation, the impregnated ITO nanoparticles can absorb IR radiation, which is effectively transferred as thermal energy to any moisture present on the fabric. This transfer of thermal energy facilitates the enhanced evaporation of moisture from the IRANISH fabric surface and as per experimental findings, a $54 \pm 9\%$ increase in the intrinsic drying rate is observed for IRANISH fabrics compared with control polyester fabrics that are treated under identical conditions, but in the absence of nanoparticles. Approach developed here for improved moisture management via the incorporation of IR absorbing nanomaterials into a textile material is novel, facile, efficient and applicable at any stage of garment manufacture. Hence, it allows us to effectively

overcome the limitations faced by existing yarn-level and structural strategies for improved moisture management.

## 1. Introduction

Diaphoresis, commonly referred to as perspiration, is a key mechanism responsible for thermoregulation of the human body. However, the accumulation of perspiration on the human body leads to notable discomfort, irritation or even undue biological consequences such as the generation of bodily odour via the bacterial metabolism of perspiration. Hence, it is a stereotype expectation that a garment, specifically those hydrophilic, will absorb and transport the perspiration, thus relieving the wearer discomfort to a considerable extent [1,2]. Nevertheless, it is notable the absorbed moisture should be promptly evaporated from the fabric surface, if wearer comfort is to persevere. The propensity of a textile material to absorb, directionally transport and evaporate moisture from its surface as described above is commonly referred to as the 'moisture management' ability of the given material. Notably, moisture management ability of a fabric and the ensuing wearer comfort are intricately related, alongside the many other undue consequences that prolonged and/or excessive moisture entrapment in a fabric may lead to. Hence, the moisture management property of a fabric is an important parameter that dictates the applicability of a given textile material in the various sectors of the garment industry.

Consequently, much recent scientific work has been focused on developing advanced materials and textiles, specifically such as those focused in novel nanotechnological innovations, with improved moisture management properties for garment applications [3–5]. Hence, nanotechnology-enabled textiles and garments have become common in the present marketplace [6]. The majority of existing products, however, adopt modification at the yarn level, where improved moisture management is achieved via the incorporation of a hydrophobic nanofibrous material onto conventional textile yarn [7], creating nanoscale porosity in textile structures [8,9] or the construction of nanoscale features on the yarn itself [10], etc. Such modifications at the yarn level do not allow these technologies to be readily applied during the latter stages of garment manufacture, hence limiting their applicability in industry as well as the ability to be customized according to the type of fabric, type of application or the many other specific requirements a given garment application would hold.

Alternatively, many recent advancements in moisture management have focused on developing generic technological solutions that are applicable at any stage of the garment industry. In their recent work, Babar *et al.* [11] reported the development of a cellulose acetate-based dual-layer nanofibre membrane with exceptional directional moisture transport as a potential substrate for fast sweat release applications. Additionally, Gao *et al.* [12] have demonstrated the development of personal thermal regulation textiles using thermally conductive and highly aligned boron nitride/poly(vinyl alcohol) composite nanofibres fabricated through a three-dimensional (3D) printing method. Similar materials have also been developed via electrospun double-layered textiles of nanofibrous poly(vinylidene fluoride) and nylon membranes as the inner hydrophobic and outer hydrophilic layers, respectively [13]. More recently, a functional textile with a trilayered fibrous membrane with elevated one-way moisture transport index in one direction and a high breakthrough pressure in the reverse was developed by introducing a transfer layer capable of guiding directional, continuous and spontaneous moisture transport through the textile [14].

Such innovations are collectively focused on developing technologies with superior wearer comfort through enhanced moisture management properties. The scalability of these technologies to an industrially viable level, however, is questionable as the successful application in industry would require mass-scale production, low cost as well as minimal interruption of conventional textile production processes. Additionally, the anticipated moisture management properties in these technologies rely on the imparted improvement in wicking properties, which would ensure the fast transport of moisture to the outer surface of a fabric. However, if wearer comfort is to persist, it is notable that the transported moisture would have to evaporate from the outer textile surface at an elevated rate, without any elevated tendency to accumulate on the fabric surface. Hence, the applicability of these technologies will be limited, specifically in warm and humid conditions, under which perspiration is a characteristic problem. Hence, novel and alternative strategies that ensure stimulated and efficient outer surface evaporation of moisture are warranted to efficiently improve the moisture management properties of textiles or garments to an adequately applicable level.

Here, the development of an infrared absorbing nanoparticle impregnated self-heating (IRANISH) fabric with enhanced moisture management properties is reported. The material is developed by incorporating ITO nanoparticles into a polyester fabric through a facile and scalable high-pressure dyeing approach. Notably, perspiration is closely linked to conditions under which solar radiation is abundant. The ITO nanoparticles impregnated into the fabric herein can efficiently absorb IR radiation which is then transferred as thermal energy onto any moisture present on the fabric surface [15]. This transfer of energy facilitates the enhanced evaporation of moisture from the IRANISH fabric surface and as per experimental findings, a $54 \pm 9\%$ increase in the intrinsic drying rate is observed for IRANISH fabrics compared with control polyester fabrics that are treated under identical conditions, but in the absence of nanoparticles.

# 2. Material and methods

## 2.1. Materials

Tin-doped indium oxide (ITO) nanoparticles were purchased from Changsha Huazun Ceramic Material Co., Ltd, China. Ultrapure water was obtained from an Evoqua Water Technologies ultrapure water system and was used in all experiments. Black-coloured, single jersey knit fabrics with a 100% polyester content and a weight of $140 \, \text{g m}^{-2}$ were used in all experiments as the specimen for surface modification.

## 2.2. Characterization of ITO

Optical properties of ITO were characterized using a Shimadzu UV-3600 UV–VIS–NIR spectrophotometer. Particle size was analysed using a Malvern Zetasizer Nano-ZS particle size analyser, while transmission electron microscopic (TEM) images obtained using a JEOL JEM-2100 high-resolution transmission electron microscope were used for the further investigation of particle size and morphology. X-ray diffraction pattern of ITO was collected using a Bruker D8 Focus powder X-ray diffractometer.

## 2.3. Fabrication of the ITO-coated polyester fabrics

The fabrics were cut into $15 \times 15$ cm pieces for use in the nanoparticle coating process. ITO nanoparticles were suspended in ultrapure water via sonication at a 2% weight percentage. Then, the fabric samples were combined with the as-prepared ITO suspension at a $100\times$ liquor ratio (i.e. the weight ratio of fabric samples : dying solution was $1 : 100$) in a commercial pressurizable container, and was allowed to dye under high-pressure boiling conditions for 45 min, hence allowing the ITO nanoparticles to diffuse into the fabric structure. The dyed samples were then allowed to dry at 120°C for 5 min to evaporate any moisture absorbed into the fabric samples, and were used as needed in the experiments discussed herein. Control samples of the same fabric were placed in the pressurizable container, allowed to stand under high-pressure boiling conditions for 45 min in water in the absence of ITO, dried at 120°C for 5 min and were used as the control samples as needed.

## 2.4. Characterization of the ITO-coated polyester fabrics

The loading of ITO nanoparticles on the IRANISH fabrics was determined by measuring the weights of $4 \times 4$ cm swatches of the IRANISH and control fabric samples ($n = 7$). Prior to measurements, all samples were conditioned for 24 h under ambient conditions (25°C, 70% relative humidity), and then the weights of the individual fabric swatches were recorded using an analytical balance. ITO-coated fabrics prepared above were imaged using a Hitachi SU6600 scanning electron microscope (SEM), and energy-dispersive X-ray (EDX) spectroscopy was conducted to identify the presence of indium using an Oxford Instruments x-act spectrometer. Atomic force microscopy (AFM) was used to characterize the surface of fabric specimens using a Park Systems XE-100 atomic force microscope, and the optical properties of fabric specimens were characterized using a Shimadzu UV-3600 UV–VIS–NIR spectrophotometer.

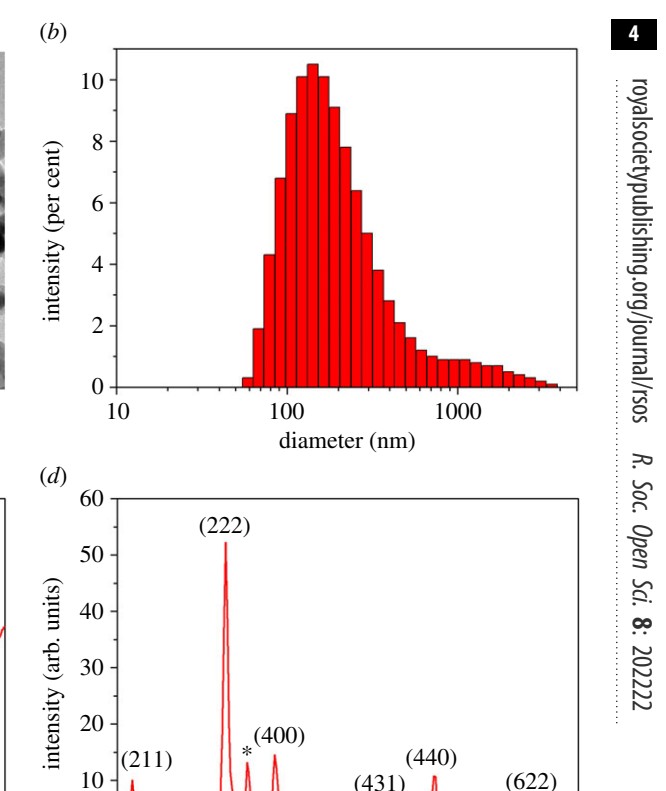

**Figure 1.** (*a*) TEM image, (*b*) particle size distribution, (*c*) UV–visible–NIR spectrum and (*d*) XRD pattern obtained for ITO nanoparticles.

## 2.5. Measurement of drying rate of fabrics

The drying rate of fabrics were determined according to a modified version of the American association for textile chemists and colorists (AATCC) test method 201–2013, drying rate of fabrics: heated plate method. Here, a custom-built apparatus comprising a ceramic heating plate that mimics body temperature (37°C) was used, while a horizontal air flow was allowed over the surface of the fabric specimen equilibrated on the heated ceramic plate. The test method was modified to include a light source which was calibrated to have a light irradiance of 1000 W m$^{-2}$ measured at the surface of the ceramic plate, by adjustment of the vertical height between the ceramic plate and the light source. The test determines the drying rate of the fabric, exposed to a known volume of water simulating human perspiration, while being in contact with a heated ceramic plate set at 37°C, the normal human body temperature. Briefly, a test specimen was first allowed to equilibrate at 37°C on the ceramic plate for 5 min. Then, the floodlight was turned on allowing the sample to equilibrate for a further 5 min under simulated solar irradiation, and 200 µl of water was introduced between the fabric and the ceramic plate, to simulate human perspiration, while making minimal disturbance to the equilibrated sample. Wicking of water through the fabric sample, the subsequent evaporation process and the temporal variation in fabric surface temperature were observed with the help of an FLIR i7 infrared (IR) thermal imaging camera.

## 3. Results

Characterization data for ITO nanoparticles are indicated in figure 1. Accordingly, these nanoparticles were prolate spheroidal in shape as seen in figure 1*a*, with an average dimension of 162.8 nm along the symmetry axis, whereas the dimensions of the particles along the other directions are less than 100 nm. The particle size distribution for ITO nanoparticles, obtained by analysis of the microscopic data, is indicated in figure 1*b*. The visible–near IR spectrum of ITO shown in figure 1*c* indicates the profound absorption

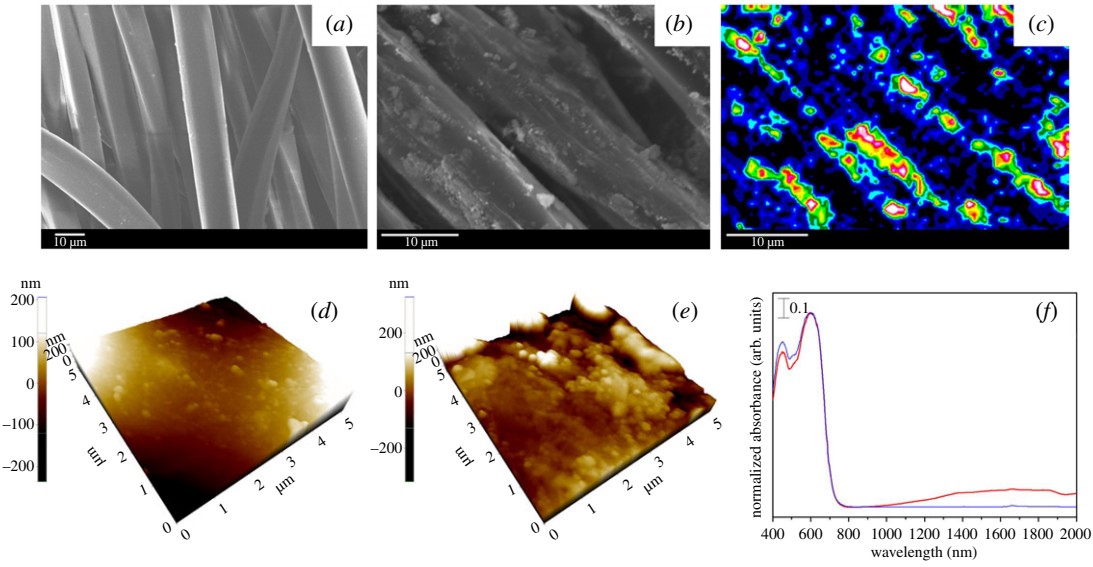

**Figure 2.** SEM images of (*a*) control and (*b*) IRANISH fabric specimens, (*c*) the EDX spectroscopic elemental map for indium of the IRANISH specimen observed in (*b*) indicating the presence of an ITO coating on the fabric surface, AFM images of (*d*) control and (*e*) IRANISH specimens, and (*f*) UV–visible–NIR spectra of IRANISH (red) and control (blue) fabric specimens.

characteristics of ITO, specifically in the near IR region. Similar absorption characteristics of ITO have been previously reported and used in various applications such as energy conversion and storage [16], sensing [17,18], localized surface plasmon resonance [19,20], photovoltaics [21], biomolecular detection [22] as well as textile-based applications [23,24] etc. Figure 1*d* shows the XRD pattern obtained for ITO nanoparticles, indicating the crystallographic features labelled on the diagram, which are in good agreement with those previously reported in the literature [19,25]. Notably, the additional peak observed at approximately 33° indicates the presence of undoped Sn as per previous reports [26,27].

The material developed here consists of a coating of ITO nanoparticles on the fabric/yarn surface at a pick-up of $1.2 \pm 0.2\%$. This can be clearly seen by the comparison of the SEM images shown in figure 2*a,b*, where a prominent coating of ITO nanoparticles is observed on the IRANISH fabric surface. This is evidenced once again in the energy-dispersive X-ray (EDX) spectroscopic elemental map for indium shown in figure 2*c*, indicating the localized presence of ITO on the yarn structures. Further, the AFM images of ITO-coated and control fabric specimens shown in figure 2*d,e* indicate the presence of an overall materials coating, resulting in a nanoscale surface topology on the otherwise smooth yarn surface. Note that the 'control' specimens used throughout this study are polyester fabrics treated under identical conditions, but in the absence of nanoparticles. Similar inference was arrived at using the X-ray photoelectron spectroscopic (XPS) analysis of IRANISH and control fabric specimens, where the corresponding spectral evidence clearly indicated the presence of ITO on the polyester fabric surface post-modification (see electronic supplementary material).

The presence of ITO nanoparticles on the fabric allows it to be competent in absorbing the IR radiation upon solar irradiation, as experimentally evidenced by the ultraviolet (UV)–visible–NIR spectra of ITO-coated (IRANISH) and control fabric specimens indicated in figure 2*f*. Thus, in order to experimentally validate the hypothesis of localized surface heating, IRANISH and control samples were equilibrated under simulated solar radiation for 5 min in a custom-built experimental set-up as indicated in figure 3*a*, as described in the experimental section. Eventually, the surface temperatures of the samples were recorded using an IR thermal imaging camera after 5 min of exposure to simulated solar radiation. Findings indicate that the IRANISH samples increase the surface temperature to $48 \pm 1°C$ from the initial value of 37°C, during the 5 min period where they were equilibrated under simulated sunlight. In contrast, the surface temperature of the control samples increased only to $43.0 \pm 0.8°C$ under identical conditions. Hence, a significant difference in surface temperature elevation (5.1°C) was observed for IRANISH samples in comparison with control samples, validating the IR radiation-stimulated self-heating property of the IRANISH fabrics.

In order to understand the above process further, the surface temperatures under simulated sunlight of an IRANISH fabric sample and a control were recorded continuously using an IR thermal imaging camera at 1 min intervals under simulated solar radiation, and the results obtained are indicated in

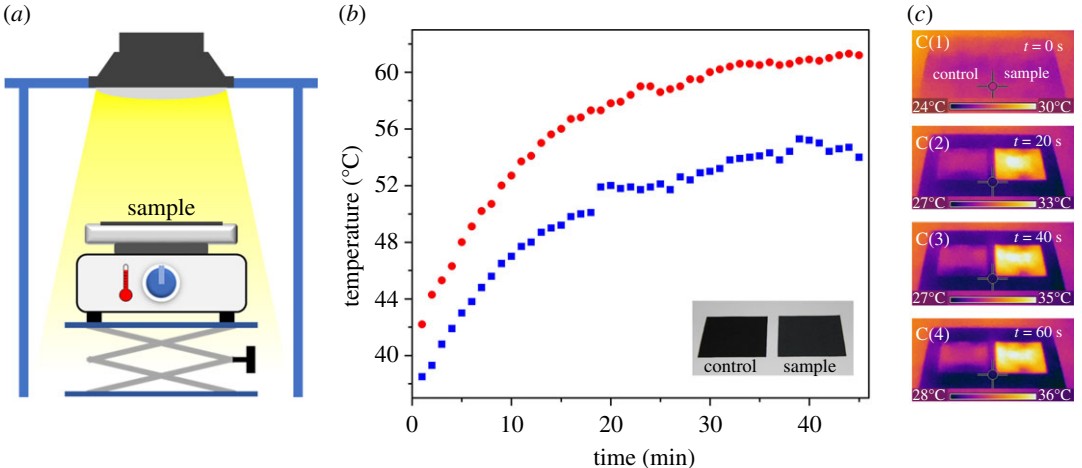

**Figure 3.** (*a*) Experimental set-up for the measurement of drying rate of fabrics consisting of a sun simulating light source and a height-adjustable ceramic hot plate, equilibrated at 37°C, on which the analysis specimen is placed, (*b*) variation of the surface temperature with time of an IRANISH fabric sample (red circles) and a control (blue squares) recorded using an IR thermal imaging camera (inset shows a photograph of the two fabric specimens under diffuse light), and (*c*) thermal images of IRANISH (right) and control (left) fabric samples kept side-by-side and captured (1) 0 s, (2) 20 s, (3) 40 s and (4) 60 s after exposure to simulated solar radiation. Note that the brighter colours in the thermal images correspond to higher temperatures.

figure 3*b*. As can be seen, the surface temperature of both fabric specimens increases upon the exposure to simulated solar radiation. The rate of increase in temperature, as well as the saturation surface temperature, however, is significantly higher in the IRANISH fabric sample compared with the control. This result, therefore, provides clear evidence for the hypothesis on IR absorption via the ITO nanoparticle impregnation and the ensuing elevation in surface temperature in the IRANISH fabric.

It is important to note that there are no notable differences in the physical appearance of the IRANISH and control fabrics as seen in the inset of figure 3*b*, regardless of their distinctively different behaviour upon the absorption of solar radiation. Additionally, no significant differences were observed in terms of the hand-feel (i.e. the sensation upon physical contact) and other prominent aesthetic properties in the IRANISH fabric compared with control specimens. It is likely that the smaller dimensions of the ITO particles as well as the nanoscale surface roughness of the coating are in combination responsible for this unaltered aesthetic effect. Nevertheless, the physical appearance and the aesthetic properties of the IRANISH fabrics are salient considerations in dictating the industrial adoptability of the IRANISH fabric technology, considering its unique ability to minimally intrude the intrinsic properties of the base fabric material.

Thermal images of an IRANISH fabric sample and a control fabric kept side-by-side on the experimental set-up are shown in figure 3*c*. The images here were acquired right before illumination (i.e. 0 s) and 20, 40 and 60 s after exposure to simulated solar radiation. As visible, there is an instantaneous and distinct elevation in the surface temperature of the IRANISH fabric sample as suggested from the previous experimental evidence. The significance of the surface heating in the IRANISH specimen here is clearly visible from the bright colours in the thermal images as is visible on the IRANISH fabric sample (extreme right), almost instantaneously after the exposure to simulated solar light. Therefore, collectively these results indicate that the material coating developed on the polyester fabric is capable of absorbing IR radiation, thus creating an IR-stimulated 'self-heating' effect on the fabric surface.

The subsequent hypothesis of the study was the facile evaporation of water from the IRANISH fabric surface as schematically illustrated in figure 4*a*. In order to investigate this effect, the drying rate of fabrics was measured as described in detail in the experimental section. Accordingly, the surface temperature of the samples was recorded at 10 s intervals after the exposure to simulated solar radiation and the results obtained are plotted in figure 4*b*. As can be seen, similar to previous results, the surface heating effect is more pronounced in the IRANISH sample compared with the control during the initial stage, before any water is introduced onto the fabric. The surface temperature of both samples, however, instantaneously decrease after introducing the 200 µl of water between the fabric sample and the ceramic plate, to simulate human perspiration (i.e. at $t = t_0 = 300$ s), indicating the efficient transfer of thermal energy to the water that is in intimate contact with the fabric, hence lowering the surface temperature.

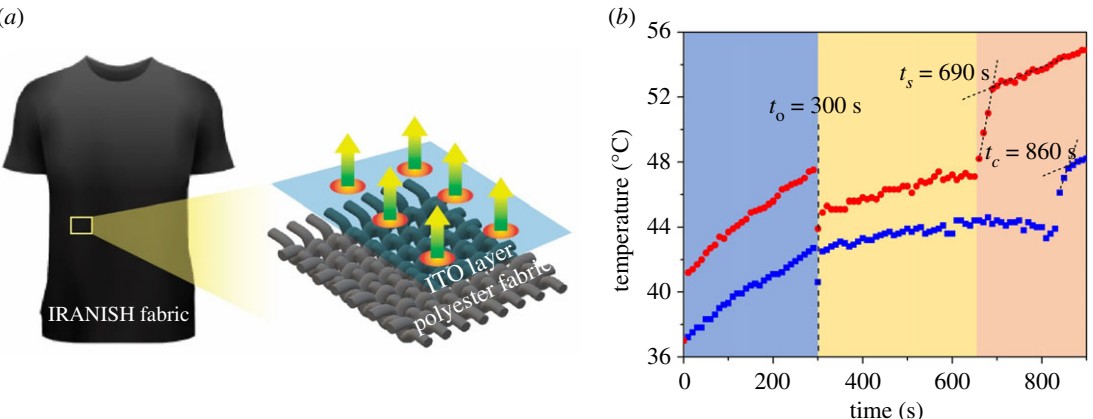

**Figure 4.** (a) Schematic illustration of the structure and function of the IRANISH fabric, indicating the hypothesized evaporation of moisture from the fabric surface and (b) the variation of surface temperature as a function of time, observed during the process of moisture evaporation from the surface of IRANISH (red circles) and control (blue squares) fabric specimens. Here, '$t_o$' corresponds to the point at which water was introduced to the fabric specimen.

The surface temperature of the control shortly returns to its original trend line, indicating that there is no significant 'self-heating' effect and/or any ensuing energy transfer. However, the surface temperature in the IRANISH fabric sample continues to follow a significantly lower trend line during the period in which water is present on the fabric. This result is significant as it indicates that (i) the IR energy absorbed via the ITO nanoparticles can be efficiently transferred to any moisture present on the fabric and (ii) as a result, the elevation in fabric surface temperature will be greatly diminished in the presence of moisture on the fabric. The above behaviour, therefore, is another important consideration for the real-world moisture management applications of the developed IRANISH technology as it suggests that this 'self-heating' effect will not lead to excessive heating of the fabric or garment, as hot and humid conditions are typically accompanied by the generation of perspiration under which the surface heating will be controlled as observed above, thereby not causing any discomfort to the wearer. However, given that the persistence of wearer comfort here is related to the presence of perspiration, it should be noted that this technology may not be suited under non-humid conditions and/or when the wearer may suffer from dehydration.

Once all water is evaporated from the fabric, however, the surface temperature indicates an instantaneous upsurge, hence allowing the clear identification of complete evaporation. Of note, the time taken for the complete evaporation of the water from the IRANISH fabric ($t_s$) is significantly lower than that of the control ($t_c$) as is visible in figure 4b. Additionally, it is clear that the surface temperature of the IRANISH fabric returns to its usual trend line almost instantaneously upon the complete evaporation of water, indicating that the 'self-heating' capacity of the developed technology can be efficiently restored in the absence of moisture.

# 4. Discussion

Solar radiation spans most of the electromagnetic spectrum, with prominent contributions from UV, visible and IR components. However, it is notable that the IR wavelengths are those primarily responsible for localized surface heating, via surface absorption and ensuing molecular vibrations. Hence, the nanomaterial used here was ITO, previously reported to be efficient in absorbing IR radiation [19,28–31], due to the plasmon oscillation generated by the electrons in the conduction band resulting from the doping process [32].

In this work, ITO nanoparticles were incorporated into black, single jersey knit, 100% polyester (weight—140 g m$^{-2}$) fabric samples through a facile high-pressure dyeing approach (vide supra). The focus was limited to polyester owing to its inherent limitations in terms of moisture management properties, as well as due to its known tendency to create user discomfort in many archetypal contexts in which polyester-based garments are used; specifically in active wear, regardless of the superior ability of polyester fabrics to transport moisture through and along the fabric structure.

It was anticipated that the high temperature used during the dyeing process would provide the ITO nanoparticles sufficient kinetic energy to penetrate into the internal crevices of the fabric, hence

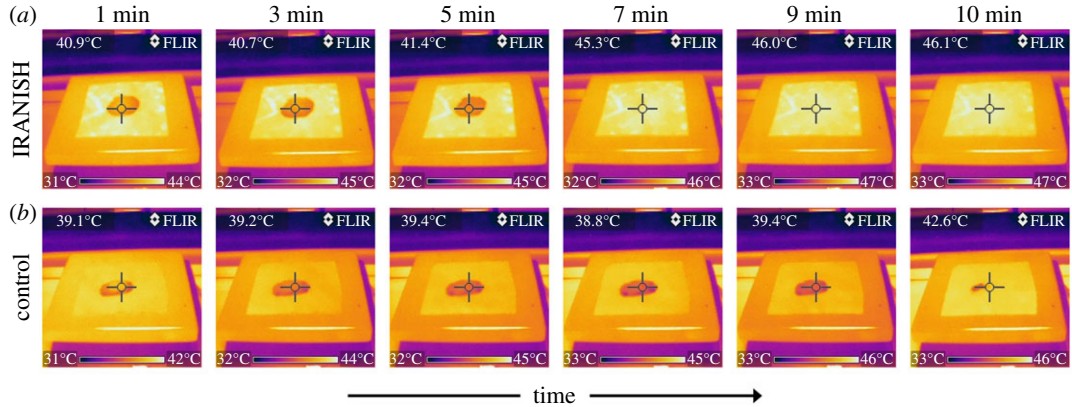

**Figure 5.** Thermal images acquired during the evaporation of water from an IRANISH (*a*) and a control (*b*) fabric samples under experimental conditions used for the measurement of drying rate of fabrics. The images in each panel have been captured 1, 3, 5, 7, 9 and 10 min, respectively, after introducing water.

facilitating the impregnation process [33]. Notably, this simple approach for the incorporation of nanoparticles onto the fabric surface is expected to aid the industrial scalability of the technology developed herein, specifically since similar high-pressure dyeing processes are commonplace in the textile industry as a fabric coloration approach.

In developing this technology, it was initially hypothesized that the presence of ITO on the surface of the fabric would lead to a spontaneous increase in the surface temperature upon exposure to solar radiation, given the propensity of ITO nanoparticles to adsorb IR radiation potentially leading to localized heating (vide infra). Hence, the material developed here is referred to as an IRANISH fabric. Next, it was hypothesized that in the presence of moisture, the IR energy absorbed by the impregnated nanoparticles can be efficiently transferred to surface moisture, hence promoting the evaporation of water from the IRANISH fabric surface. Thus, the developed technology was anticipated to facilitate the facile evaporation of perspiration from garments leading to effective and improved moisture management.

In order to further quantify the improvement in moisture management properties of the IRANISH fabric, the improvement in drying rate of the IRANISH fabrics with respect to the controls was calculated according to the equation given below

$$\text{improvement in drying rate} = \left( \frac{\text{drying time of control specimen}}{\text{drying time of IRANISH specimen}} - 1 \right) \times 100\%. \tag{4.1}$$

(see electronic supplementary material for the derivation of equation (4.1)). As an example, for the data shown in figure 4*b*, the drying time of the IRANISH specimen (i.e. $t_s - t_0$) is 390 s, whereas that of the control (i.e. $t_c - t_0$) is 560 s. Hence, according to equation (4.1), the improvement in the drying rate of the IRANISH sample is 44%, with respect to the control. From the repeated analysis conducted herein, it was confirmed that on average the nanoparticle impregnated fabrics indicate a $54 \pm 9\%$ improvement in the drying rate with respect to the control, indicating a significant improvement in the drying rate of the IRANISH fabrics.

To visualize this phenomenon, thermal images of IRANISH and control specimens were captured at predefined time intervals under experimental conditions used for the measurement of drying rate of fabrics discussed above. The thermal images obtained at 1, 3, 5, 7, 9 and 10 min, respectively, after introducing water for both IRANISH and control specimens are shown in figure 5. These thermal images reconfirm that the surface temperature is higher in the IRANISH specimen, in comparison with the control specimen at each comparable time duration, hence indicating the 'self-heating' capacity of the IRANISH fabric. Additionally, although the vertical wicking of water is almost instantaneous in both fabrics, thereby allowing the water to reach the top of the fabric surface, the droplet is spread in a comparable or perhaps even a larger surface area in the IRANISH fabric compared with the control, indicating that the intrinsic spreading properties are unaltered or possibly even improved by the modifications conducted herein. This serves as an added benefit as the process of moisture evaporation is intimately supported by the vertical moisture transport through a fabric or garment as well as the ensuing efficient surface spreading of moisture.

Importantly, as visible from the thermal images, it is clear that for the IRANISH specimen, complete evaporation of water is observed within 5–7 min of introducing water, whereas the surface temperature of it indicates a steady increase beyond this point. By contrast, however, trace amounts of water are clearly observed on the control specimen even 10 min after introducing water into the experiment. Hence, it is clear that the energy transfer facilitated in the IRANISH fabric overcomes the subtle variations in the surface spreading of moisture observed between the IRANISH and control specimens, thus becoming dominant in creating a prominent 'quick-dry' effect in the IRANISH fabric as quantitatively determined above.

Additionally, it is notable that the surface temperature of both IRANISH and control specimens does not indicate any distinct variations at the vicinity of the water droplet as observed for the thermal images given in figure 5 (note the temperature in each case is measured at the centre of the cursor). The surface temperature of the IRANISH specimen, however, indicates a distinct upsurge once all water is evaporated from the fabric surface, as seen by the temperatures recorded for the IRANISH fabrics in the thermal images acquired at 7 or more minutes after introducing water. This fact reassures that the elevation in surface temperature, or the 'self-heating', will be greatly diminished in the presence of moisture on the fabric surface, thereby not causing any discomfort to the wearer under conditions where perspiration is apparent, indicating the perseverance of wearer comfort via this novel technology.

## 5. Conclusion

Here, the development of a nanoparticle impregnated self-heating fabric, named as an IRANISH fabric, with enhanced moisture management properties is reported. Experimental findings have confirmed that a $54 \pm 9\%$ increase in the intrinsic drying rate is observed for IRANISH fabrics compared with control polyester fabrics that are treated under identical conditions, but in the absence of nanoparticles. Of note, the ITO nanoparticles are impregnated into the fabric through a facile high-pressure dyeing approach, indicating that the process could be easily adopted at any stage of garment manufacture; an important consideration dictating the industrial scalability and the practical viability of the developed technology. Thus, overall, the technology developed here is novel, facile, efficient, industrially scalable and commercially viable, and hence, it allows to overcome the limitations faced by the existing strategies for improving the moisture management properties of textiles. Interestingly, although perspiration is closely linked to the presence of solar radiation, there are no previous reports on the use of solar radiation as an external stimuli, as well as a driving force, for improved moisture management in textiles. This novel technology, therefore, holds great promise in creating garments with significantly improved drying rates, specifically applicable for the moisture management under hot and humid conditions at which perspiration is a notable concern, thus demanding novel and improved moisture management strategies as that developed herein.

Data accessibility. Some data supporting this article have been uploaded as part of the electronic supplementary material. In addition, the raw datasets can be found in the Dryad Digital Repository: https://doi.org/10.5061/dryad.905qfttjs [34].
Authors' contributions. L.A.W. and W.R.L.N.B. carried out most of the experiments and wrote the first draft of the paper. R.N.W. and N.D.T. contributed towards the expertise in textile engineering and with characterization. G.J.A. and K.M.N.D.S. supervised the project and fine-tuned the final manuscript. All authors discussed the results and commented on the final manuscript.
Competing interests. We have no competing interests.
Funding. The authors would like to express gratitude towards MAS Active Trading (Pvt) Ltd for the funding provided for research work and for the subsequent transfer of the technology through exclusive acquisition of intellectual property (patent publication no. US2017/0314185 A1) for prospective industrial commercialization.
Acknowledgements. We would like to thank Dr Ishara Dharmasena, Ms Nissansala Bandara, Ms Indrachapa Bandara, Mr Chanaka Sandaruwan and Ms Damayanthi Dahanayake for assisting in various stages of the project.

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
