## [Peer Review File · Royal Society Open Science]

Review History

RSOS-202222.R0 (Original submission)

Review form: Reviewer 1

Is the manuscript scientifically sound in its present form?

No

Are the interpretations and conclusions justified by the results?

No

Is the language acceptable?

Yes

Do you have any ethical concerns with this paper?

No

Have you any concerns about statistical analyses in this paper?

No

Recommendation?

Major revision is needed (please make suggestions in comments)

Comments to the Author(s)

In this manuscript, Infrared Absorbing Nanoparticles are used to convert solar radiation into heat for accelerated drying of the wetted fabrics. However, the highly increased temperature will make the wearers uncomfortable. The authors may first agree if the thermal comfort can be satisfied based on the proposed moisture management fabric.

Review form: Reviewer 2

Is the manuscript scientifically sound in its present form?

No

Are the interpretations and conclusions justified by the results?

No

Is the language acceptable?

Yes

Do you have any ethical concerns with this paper?

No

Have you any concerns about statistical analyses in this paper?

Yes

Recommendation?

Major revision is needed (please make suggestions in comments)

Comments to the Author(s)

The author describes a facile approach to impregnating Tin doped Indium Oxide (ITO) nanoparticles into a polyester fabric through a high pressure dyeing approach.

The scientific background in terms of reliable experimental data is acceptable, but the expression given in Discussion is not well supported by solid evidence. The work in this paper is needed to be improved before it can be published.

1. I doubt whether the nanoparticles can adhere firmly to the fiber, what is the force of the nanoparticle binding on the fabric?
2. As a fabric, it must be washable and can be worn repeatedly. Can these nanoparticles still adhere to the fibers of the fabric after multiple washings? Did the intrinsic drying rate still perform well after multiple washings?
3. After modification, what is the loading amount of nanoparticles on the fabric?
4. In figure 3(c) and Figure 5, the temperature scale should be given in the thermal image.
5. After modification, the FTIR and XPS should also be given to prove the ITO nanoparticles had been coated on the fabric.

Decision letter (RSOS-202222.R0)

Dear Professor Nalin De Silva:

Title: Infrared Absorbing Nanoparticle Impregnated Self-heating Fabrics for Significantly Improved Moisture Management under Ambient Conditions
Manuscript ID: RSOS-202222

The editor assigned to your manuscript has now received comments from reviewers. We would like you to revise your paper in accordance with the referee and Subject Editor suggestions which can be found below (not including confidential reports to the Editor). Please note this decision does not guarantee eventual acceptance.

Please submit your revised paper before 05-Feb-2021. Please note that the revision deadline will expire at 00.00am on this date. If we do not hear from you within this time then it will be assumed that the paper has been withdrawn. In exceptional circumstances, extensions may be possible if agreed with the Editorial Office in advance. We do not allow multiple rounds of revision so we urge you to make every effort to fully address all of the comments at this stage. If deemed necessary by the Editors, your manuscript will be sent back to one or more of the original reviewers for assessment. If the original reviewers are not available we may invite new reviewers.

Royal Society of Chemistry
Thomas Graham House
Science Park, Milton Road
Cambridge, CB4 0WF

Royal Society Open Science - Chemistry Editorial Office

On behalf of the Subject Editor Professor Anthony Stace and the Associate Editor Professor Chaohua Cui.

RSC Associate Editor: 1
Comments to the Author:
(There are no comments.)

RSC Associate Editor: 2
Comments to the Author:
(There are no comments.)

Reviewers' Comments to Author:
Reviewer: 1

Comments to the Author(s)

In this manuscript, Infrared Absorbing Nanoparticles are used to convert solar radiation into heat for accelerated drying of the wetted fabrics. However, the highly increased temperature will make the wearers uncomfortable. The authors may first agree if the thermal comfort can be satisfied based on the proposed moisture management fabric.

Reviewer: 2

Comments to the Author(s)

The author describes a facile approach to impregnating Tin doped Indium Oxide (ITO) nanoparticles into a polyester fabric through a high pressure dyeing approach. The scientific background in terms of reliable experimental data is acceptable, but the expression given in Discussion is not well supported by solid evidence. The work in this paper is needed to be improved before it can be published.

1. I doubt whether the nanoparticles can adhere firmly to the fiber, what is the force of the nanoparticle binding on the fabric?
2. As a fabric, it must be washable and can be worn repeatedly. Can these nanoparticles still adhere to the fibers of the fabric after multiple washings? Did the intrinsic drying rate still perform well after multiple washings?
3. After modification, what is the loading amount of nanoparticles on the fabric?
4. In figure 3(c) and Figure 5, the temperature scale should be given in the thermal image.
5. After modification, the FTIR and XPS should also be given to prove the ITO nanoparticles had been coated on the fabric.

Author's Response to Decision Letter for (RSOS-202222.R0)

See Appendix A.

RSOS-202222.R1 (Revision)

Review form: Reviewer 1

Is the manuscript scientifically sound in its present form?

Yes

Are the interpretations and conclusions justified by the results?

No

Is the language acceptable?

Yes

Do you have any ethical concerns with this paper?

No

Have you any concerns about statistical analyses in this paper?

No

Recommendation?

Accept with minor revision (please list in comments)

Comments to the Author(s)

The paper has been improved. However, the fabric temperature still looks high. The authors may conduct a simple human trial test and evaluate the comfort property.

The wearers must sweat, or they will feel very hot. What about when the wearers suffer from dehydration? Perhaps the limitations of this research should be discussed.

In the literature, the following references of thermal and moisture management are related, e.g., Composites Communications, 2021, 100595; Advanced Functional Materials, 2018, 28 (36), 1800269; Applied Physics Letters, 2014, 104 (23), 231602.

Review form: Reviewer 2

Is the manuscript scientifically sound in its present form?

Yes

Are the interpretations and conclusions justified by the results?

Yes

Is the language acceptable?

Yes

Do you have any ethical concerns with this paper?

No

Have you any concerns about statistical analyses in this paper?

No

Recommendation?

Major revision is needed (please make suggestions in comments)

Comments to the Author(s)

The process of dyeing and loading nanoparticles are different, the reason that high temperature high pressure (HTHP) method used in dyeing was because HTHP can help dye molecules enter the amorphous areas inside the fiber. However, the nanoparticles are too large to get inside the fibers. Those ITO nanoparticles only physically trap within the internal crevices of the fabric structure. So the adhesion fastness of nanoparticles are doubtful. And you still did not prove that Whether the nanoparticles can still stick to the fabric after being washed many times.

Decision letter (RSOS-202222.R1)

Dear Professor Nalin De Silva:

Title: Infrared Absorbing Nanoparticle Impregnated Self-heating Fabrics for Significantly Improved Moisture Management under Ambient Conditions
Manuscript ID: RSOS-202222.R1

The editor assigned to your paper has now received comments from reviewers. We would like you to revise your paper in accordance with the referee and Subject Editor suggestions which can be found below (not including confidential reports to the Editor). Please note this decision does not guarantee eventual acceptance.

Please submit a copy of your revised paper before 04-Apr-2021. Please note that the revision deadline will expire at 00.00am on this date. If we do not hear from you within this time then it will be assumed that the paper has been withdrawn. In exceptional circumstances, extensions may be possible if agreed with the Editorial Office in advance. We do not allow multiple rounds of revision so we urge you to make every effort to fully address all of the comments at this stage. If deemed necessary by the Editors, your manuscript will be sent back to one or more of the original reviewers for assessment. If the original reviewers are not available we may invite new reviewers.

When submitting your revised manuscript, you must respond to the comments made by the referees and upload a file "Response to Referees" in "Section 6 - File Upload". Please use this to document how you have responded to the comments, and the adjustments you have made. In

order to expedite the processing of the revised manuscript, please be as specific as possible in your response.

On behalf of the Subject Editor Professor Anthony Stace and the Associate Editor Professor Chaohua Cui.

RSC Associate Editor:
Comments to the Author:
(There are no comments.)

RSC Subject Editor:
Comments to the Author:
(There are no comments.)

Reviewers' Comments to Author:
Reviewer: 2

Comments to the Author(s)

The process of dying and loading nanoparticles are different, the reason that high temperature high pressure (HTHP) method used in dying was because HTHP can help dye molecules enter the amorphous areas inside the fiber. However, the nanoparticles are too large to get inside the fibers. Those ITO nanoparticles only physically trap within the internal crevices of the fabric structure. So the adhesion fastness of nanoparticles are doubtful. And you still did not prove that Whether the nanoparticles can still stick to the fabric after being washed many times.

Reviewer: 1

Comments to the Author(s)

The paper has been improved. However, the fabric temperature still looks high. The authors may conduct a simple human trial test and evaluate the comfort property.

The wearers must sweat, or they will feel very hot. What about when the wearers suffer from dehydration? Perhaps the limitations of this research should be discussed.

In the literature, the following references of thermal and moisture management are related, e.g., Composites Communications, 2021, 100595; Advanced Functional Materials, 2018, 28 (36), 1800269; Applied Physics Letters, 2014, 104 (23), 231602.

Author's Response to Decision Letter for (RSOS-202222.R1)

See Appendix B.

RSOS-202222.R2 (Revision)

Review form: Reviewer 1

Is the manuscript scientifically sound in its present form?

Yes

Are the interpretations and conclusions justified by the results?

Yes

Is the language acceptable?

Yes

Do you have any ethical concerns with this paper?

No

Have you any concerns about statistical analyses in this paper?

No

Recommendation?

Accept as is

Comments to the Author(s)

The manuscript can be accepted.

Review form: Reviewer 2

Is the manuscript scientifically sound in its present form?

Yes

Are the interpretations and conclusions justified by the results?

Yes

Is the language acceptable?

Yes

Do you have any ethical concerns with this paper?

No

Have you any concerns about statistical analyses in this paper?

No

Recommendation?

Accept as is

Comments to the Author(s)

could be accept

Decision letter (RSOS-202222.R2)

Dear Professor Nalin De Silva:

Title: Infrared Absorbing Nanoparticle Impregnated Self-heating Fabrics for Significantly Improved Moisture Management under Ambient Conditions
Manuscript ID: RSOS-202222.R2

It is a pleasure to accept your manuscript in its current form for publication in Royal Society Open Science. The chemistry content of Royal Society Open Science is published in collaboration with the Royal Society of Chemistry.

On behalf of the Subject Editor Professor Anthony Stace and the Associate Editor Professor Chaohua Cui.

RSC Associate Editor:
Comments to the Author:
(There are no comments.)

RSC Subject Editor:
Comments to the Author:
(There are no comments.)

Reviewer(s)' Comments to Author:
Reviewer: 1

Comments to the Author(s)
The manuscript can be accepted.

Reviewer: 2

Comments to the Author(s)
could be accept

Appendix A

Responses to reviewers' comments

Infrared Absorbing Nanoparticle Impregnated Self-heating Fabrics for Significantly Improved Moisture Management under Ambient Conditions

Manuscript ID: RSOS-202222

Reviewer: 1

Comment	Response
In this manuscript, Infrared Absorbing Nanoparticles are used to convert solar radiation into heat for accelerated drying of the wetted fabrics. However, the highly increased temperature will make the wearers uncomfortable. The authors may first agree if the thermal comfort can be satisfied based on the proposed moisture management fabric.	We are grateful for your positive feedback and the comments provided for the improvement of our manuscript. As per your question, the mechanism of enhancing the rate of moisture evaporation here is based on localized heating produced by the absorption of infrared radiation. This is depicted by the thermal images of an IRANISH fabric sample and a control fabric kept side-by-side on the experimental setup as shown in Figure 3 (C) of the manuscript. It is observed that under simulated solar radiation, the impregnated ITO nanoparticles can absorb IR radiation, which is effectively transferred as thermal energy to any moisture present on the fabric. However, under hot and humid conditions, perspiration is likely to result, and under such conditions, the IR energy absorbed by the IRANISH fabric will be efficiently transferred to the moisture (perspiration), thus relieving the fabric of any ensuing elevation in surface temperature. Hence, we anticipate minimal wearer discomfort under hot and humid conditions under which the use of this technology is advocated. This fact had been discussed in the manuscript by the experimental evidence presented in Figure 4 of the manuscript. As we state, "to visualize this phenomenon, thermal images of IRANISH and control specimens were captured at predefined time intervals under experimental conditions used for the measurement of drying rate of fabrics discussed above. The thermal images obtained at 1, 3, 5, 7, 9, and 10 minutes respectively after introducing water for both IRANISH and control specimens are shown in Figure 5". As can be seen by the thermal images acquired for IRANISH and control specimens 1 min after introducing water (left most panels in figure 5), the elevation in temperature of the IRANISH fabric is less than 2 °C compared to

	the control in the presence of moisture (40.9 vs 39.1 °C). This behavior seems to continue as long as there is moisture remaining on the fabric (please see the thermal images obtained at 1, 3, and 5 minutes). Hence, as we state in the manuscript “‘self-heating’ effect will not lead to excessive heating of the fabric or garment, as hot and humid conditions are typically accompanied by the generation of perspiration under which the surface heating will be controlled as observed above, thereby not causing any discomfort to the wearer”.
--	---

Reviewer: 2

Comment	Response
I doubt whether the nanoparticles can adhere firmly to the fiber, what is the force of the nanoparticle binding on the fabric?	We are grateful for your positive feedback and the comments provided for the improvement of our manuscript. In the reported work, the material is developed by incorporating ITO nanoparticles into a polyester fabric through a facile and scalable high-pressure dyeing approach. According to the textile industry, the most common fabric dyeing technique is high temperature high pressure (HTHP) dyeing method. Hence, the above technique was considered in our work due to the scalability and industrial applicability of the developed material. Typically, as we state in the paper, the high temperature and pressure used during the dyeing process would provide the ITO nanoparticles sufficient kinetic energy to penetrate into the internal crevices of the fabric, hence facilitating the impregnation process (D. Patterson, R.P. Sheldon, Transactions of the Faraday Society, 55 (1959) 1254-1264.). Similarly, in our work, the dyeing process was conducted in a commercial pressurizable container, and the fabrics were allowed to dye with ITO nanoparticles under high-pressure boiling conditions for 45 minutes, hence allowing the ITO nanoparticles to diffuse into the fabric structure. This is akin to what is typically adopted in industry and the high temperature and pressure would allow the ITO nanoparticles to diffuse through and physically trap within the internal crevices of the fabric

	structure. Nevertheless, once the nanoparticles diffuse into the structure of the fabric, in comparison to forming adsorption multilayers on the external fabric surface, it is likely that increased degree of intermolecular interactions may ensue between ITO and polyester thereby binding them together, thereby allowing ITO nanoparticles to adhere firmly on the polyester fabric.
As a fabric, it must be washable and can be worn repeatedly. Can these nanoparticles still adhere to the fibers of the fabric after multiple washings? Did the intrinsic drying rate still perform well after multiple washings?	Dyestuff included onto a fabric using the HTHP dyeing method have been reported to show minimal ability to leach into washings due to the strong entrapment within the fabric structures and extent of interactions that exist between the dye material and the fabric. It is likely that the high temperature and pressure used during the dyeing process would provide the ITO nanoparticles sufficient kinetic energy to penetrate into the internal crevices of the fabric, hence facilitating the impregnation process. As indicated above, we anticipate strong interactions to exist between ITO and the fabric given that almost all ITO that was used in the liquor solution was absorbed into the fabric during the dyeing process, as evidenced by the complete discoloration of the originally bluish liquor solution used for dyeing. Given that ITO has a distinct blue color, it would be apparent upon release into solution. Notably, we observe similar behavior with our material as once the material is dried (i.e. fixations) after the dyeing process, the amount of material leaching into the water was minimal. Hence, we believe that the developed method would produce wash fastness for the self-heating activity to prevail for adequate number of wash cycles.
After modification, what is the loading amount of nanoparticles on the fabric?	We appreciate your concern in this regard and the inclusion of the suggested information will certainly improve the quality of presentation and further strengthen our argument. Accordingly, we have determined the loading of ITO on the polyester fabric to be 1.2 ± 0.2 %. The determination was conducted by measuring the weights of fabric swatches of 4 x 4 cm from the IRANISH and control fabric samples ($n = 7$). Prior to measurements, all samples were conditioned for 24 hours under ambient conditions (25 °C, 70%

	relative humidity), and the weights of the individual fabric swatches were recorded using an analytical balance.
In figure 3(c) and Figure 5, the temperature scale should be given in the thermal image.	We appreciate your concern in this regard and the inclusion of the suggested information will certainly improve the quality of presentation and further strengthen our argument. Figure 3 has been modified accordingly to include the temperature scales at the bottom of each thermal image. Please note that the temperature scales in have been already indicated on the bottom of each thermal image in Figure 5.
After modification, the FTIR and XPS should also be given to prove the ITO nanoparticles had been coated on the fabric.	We appreciate your concern in this regard and the inclusion of the suggested information will certainly improve the quality of presentation and further strengthen our argument. As suggested the XPS data have been acquired and incorporated into the revised manuscript. We have also collected the FTIR spectra of the IRANISH and control specimens and have included the spectra below (IRANISH - red and control - blue). However, we have not observed any peaks indicative of the binding ITO onto the polyester, likely due to the relatively small percentage ($1.2 \pm 0.2 \%$) of ITO in the IRANISH specimen. Nevertheless, although the amount of ITO on the specimens can be increased via the alteration of the liquor concentration and/or the liquor ratio used in the preparation of the IRANISH fabric, we believe that the effective moisture management activity reported in our work, occur even at such low nanoparticle loading, thus serving as an added advantage. 
Appendix B

Responses to reviewers' comments

Infrared Absorbing Nanoparticle Impregnated Self-heating Fabrics for Significantly Improved Moisture Management under Ambient Conditions

Manuscript ID: RSOS-202222

Reviewer 1

Comment	Response
The paper has been improved. However, the fabric temperature still looks high. The authors may conduct a simple human trial test and evaluate the comfort property. The wearers must sweat, or they will feel very hot. What about when the wearers suffer from dehydration? Perhaps the limitations of this research should be discussed.	Thank you for the valuable insight in this regard. We have conducted wearer trials with similar technologies where the nanoparticles were screen printed on the fabrics rather than being embedded into the structure. The results indicated that the wearers observed no discomfort due to heating. However, the testing here was conducted under warm and humid conditions, the typical conditions under which active wear is used, while the same conditions have been advocated for the use of this technology. However, we agree that this technology would indicate limitations under non-humid conditions and/or wearer dehydration. Hence, as suggested, the limitations of the developed technology were discussed in the revised manuscript.
In the literature, the following references of thermal and moisture management are related, e.g., Composites Communications, 2021, 100595; Advanced Functional Materials, 2018, 28 (36), 1800269; Applied Physics Letters, 2014, 104 (23), 231602.	Thank you for forwarding these references. They have been cited in the revised manuscript as necessary.

Reviewer 2

Comment	Response
The process of dyeing and loading nanoparticles are different, the reason that high temperature high pressure (HTHP) method used in dyeing was because HTHP can help dye molecules enter the amorphous areas inside the fiber. However, the nanoparticles are too large to get inside the fibers. Those ITO nanoparticles only physically trap within the internal crevices of the fabric structure. So the adhesion fastness of nanoparticles are doubtful. And you still did not prove that Whether	HTHP process is used for dyeing of polyester fabrics using disperse dyes [1]. These dyes are characterized by very limited solubility in water at room temperature. They are also applied from a fine aqueous dispersion at a HTHP condition, facilitating the diffusion of the dyes into the amorphous areas of the fiber. In dispersed state, disperse dyes are known to show hydrodynamic particle sizes typically over 1 μm [2]. The nanoparticles used in the present work is also

the nanoparticles can still stick to the fabric after being washed many times.

applied to the fabric as a fine aqueous dispersion through HTHP process. Therefore, it's possible to hypothesize that penetration of ITO is feasible by the current process. Also, literature indicates very similar approaches involving HTHP process has been demonstrated in obtaining wash durable textile materials with TiO₂ nanoparticles[3,4], carbon black [5] and SiO₂ [6].

To experimentally validate reviewer observations with experimental data, we have subjected the IRANISH fabric to accelerated wash according to a method similar to AATCC 61. Thus, washed fabric (3 cm x 10 cm) is then equilibrated on a ceramic hot plate maintained at 37 °C along with unwashed and untreated polyester fabrics (Fig 1). After equilibrating for 5 mins, light was turned on temperature was measured on the surface of each fabric (Fig 2). It was observed that temperature on the washed IRANISH fabric after 2 min of light exposure is comparable to the unwashed fabric (Fig 3). This indicates that the current material has a high wash durability.

Figure 1: Samples are being equilibrated on a ceramic hotplate maintained at 37°C. Samples: Left -unwashed fabric, Mid -washed fabric, Right – untreated fabric

Figure 2: Fabrics after 2 min of light exposure, Fabrics-same order

Figure 3: Thermal camera image indicating the surface temperature of the tested fabrics, Fabrics – same order

References

- [1] Aspland, J.R., 1992. Disperse dyes and their application to polyester. *Textile Chemist and Colorist*, 24, pp.18-18.
- [2] Lee, K.W. and Kim, J.P., 2001. Effect of ultrasound on disperse dye particle size. *Textile Research Journal*, 71(5), pp.395-398.
- [3] Li, Z., Dong, Y., Li, B., Wang, P., Chen, Z. and Bian, L., 2018. Creation of self-cleaning polyester fabric with TiO₂ nanoparticles via a simple exhaustion process: Conditions optimization and stain decomposition pathway. *Materials & Design*, 140, pp.366-375.

[4] Al-Etaibi, A.M. and El-Asery, M.A., 2020. Nano TiO₂ imparting multifunctional performance on dyed polyester fabrics with some disperse dyes using high temperature dyeing as an environmentally benign method. *International journal of environmental research and public health*, 17(4), p.1377.

[5] Li, D. and Sun, G., 2007. Coloration of textiles with self-dispersible carbon black nanoparticles. *Dyes and Pigments*, 72(2), pp.144-149.

[6] Patel, B.H., Patel, P.N., Chaudhari, S.B. and Mandot, A.A., 2016. Nano silica mediated sol-gel dyeing of cotton and polyester fabric. *International Dyer*, 201(3), pp.38-41.